# Burden of Disease Due to Traffic Noise in Germany

**DOI:** 10.3390/ijerph16132304

**Published:** 2019-06-28

**Authors:** Myriam Tobollik, Matthias Hintzsche, Jördis Wothge, Thomas Myck, Dietrich Plass

**Affiliations:** 1German Environment Agency, Section Exposure Assessment and Environmental Health Indicators, Corrensplatz 1, 14195 Berlin, Germany; 2Department of Environment and Health, School of Public Health, Bielefeld University, Bielefeld, Universitätsstraße 25, 33615 Bielefeld, Germany; 3German Environment Agency, Section Noise Abatement of Industrial Plants and Products, Noise Impact, Wörlitzer Platz 1, 06844 Dessau-Roßlau, Germany

**Keywords:** environmental noise, burden of disease, traffic noise, aircraft noise, road traffic noise, railway noise, disability-adjusted life year, DALY, Germany

## Abstract

Traffic noise is nearly ubiquitous and thus can affect the health of many people. Using the German noise mapping data according to the Directive 2002/49/EC of 2017 and exposure-response functions for ischemic heart disease, noise annoyance and sleep disturbance assessed by the World Health Organization’s Environmental Noise Guidelines for the European Region the burden of disease due to traffic noise is quantified. The burden of disease is expressed in disability-adjusted life years (DALYs) and its components. The highest burden was found for road traffic noise, with 75,896 DALYs when only considering moderate evidence. When including all available evidence, 176,888 DALYs can be attributable to road traffic noise. The burden due to aircraft and railway noise is lower because fewer people are exposed. Comparing the burden by health outcomes, the biggest share is due to ischemic heart disease (90%) in regard to aircraft noise, however, the lowest evidence was expressed for the association between traffic noise and ischemic heart disease. Therefore, the results should be interpreted with caution. Using alternative input parameters (e.g., exposure data) can lead to a much higher burden. Nevertheless, environmental noise is an important risk factor which leads to considerable loss of healthy life years.

## 1. Introduction

Large parts of the population in Germany are affected by harmful levels of traffic noise. It reduces the quality of life and poses a considerable risk to human health. Traffic noise from road vehicles, railways and aircraft annoys residents and may lead to negative health impacts. The health effects resulting from traffic noise are diverse. They include cardiovascular diseases, noise annoyance, sleep disturbance, cognitive impairment (especially in children), hearing impairment and tinnitus as well as psychological disorders, such as depressive episodes and diseases like diabetes or obesity [1,2,3,4,5].

The World Health Organization (WHO) recently updated its Environmental Noise Guidelines for the European Region with the aim to provide recommendations for protecting human health from exposure to environmental noise [2]. The guidelines are based on a comprehensive systematic analysis of the pertinent literature on the associations between environmental noise from five different noise sources (road traffic, railway, aircraft, wind turbines and leisure activities) and various health effects. In total, eight systematic literature reviews were conducted, covering a broad range of health effects [1,6,7,8,9,10,11,12]. The WHO defined five of the analyzed health outcomes as critical for the assessment of environmental noise impacts: cardiovascular diseases, annoyance, effects on sleep, cognitive impairment, hearing impairment and tinnitus.

To compare the different health impacts of environmental noise, it is necessary to assess the burden of disease using a standardized measure of population health. The disability-adjusted life year (DALY) measure combines mortality and morbidity effects and thus, enables to assess the health effects of different health outcomes in a comprehensive and comparable way [13,14,15]. The aim of this study is to use the current body of evidence on the adverse health effects of traffic noise to quantify the burden of disease resulting from noise attributable to road, railway and air traffic in Germany. For this purpose, the results of the German noise mapping according to the EU Environmental Noise Directive (2002/49/EC) are used to estimate the number of people exposed to the different kinds of traffic noise.

## 2. Materials and Methods 

### 2.1. Quantification Method

The burden of disease is expressed in DALYs. This measure of population health combines mortality and morbidity in a single metric quantifying the burden of disease in healthy life years lost [13,14,15]. Mortality effects are expressed in years of life lost due to premature mortality (YLLs) and quantified by multiplying the number of deaths in a certain age group by the remaining life expectancy at the age of death. For the oldest age group (90+ years) the life expectancy for the age group 90 to 94 years is used. Morbidity, expressed as years lived with disability (YLDs), is quantified by multiplying the number of prevalent disease cases or the number of people affected by a health condition with the corresponding disability weight. Disability weights are weighting factors for the severity of diseases anchored on a scale from 0 to 1. Zero is representing a status of full health whereas one is understood as a state comparable with death [16,17,18,19].

The population attributable fraction (PAF) is used to estimate the share of disease burden which is attributable to noise exposure (see Figure 1). To quantify the PAF, the number of people exposed to noise, an exposure-response function or relative risks and a counterfactual value (alternative scenario, indicating a minimum or no exposure risk level) are needed [20,21]. The relative risks often include a 95% confidence interval which is used to quantify the confidence interval for the DALYs, YLLs, and YLDs. In our case, the counterfactual value gives the starting point for the risk increase of the exposure-response function. Below this value, no negative health effects are quantified.

The burden of disease of annoyance and sleep disturbance is quantified by using the exposure-response function to quantify the percentage of people feeling highly annoyed (%HA) or highly sleep disturbed (%SD) due to certain average noise exposure. This percentage is then applied to the people exposed to different noise levels to obtain the absolute number of people feeling highly annoyed or highly sleep disturbed. In the last step, the number of people feeling highly annoyed and highly sleep disturbed is multiplied by the corresponding disability weight, resulting in YLDs. No confidence interval could be estimated for annoyance and sleep disturbance based on the exposure-response function because no confidence interval was given in the publication.

The calculations are performed in Microsoft Excel 2013. Uniform age-weights and no time-discount are applied. Where possible, the models are run for both sexes separately and for five years age groups. The last age group includes all people above the age of 90 years.

### 2.2. Health Outcomes and Exposure-Response Functions

For the update of the WHO Environmental Noise Guidelines for the European Region, several systematic literature reviews on the effects of environmental noise were conducted [2]. The considered health effects, which are seen as critical for assessing environmental noise impacts, are cardiovascular diseases, annoyance, effects on sleep, cognitive impairment, hearing impairment, and tinnitus. Noise annoyance is the subjective feeling of being disturbed, annoyed or bothered by noise. It is assessed retrospectively by two standardized questions [22]. In this assessment, high long-term annoyance is assessed, which in the following is called annoyance for reasons of simplicity. Sleep disturbance can be assessed in many different ways. For this assessment, we rely on the WHO recommendations and assess the burden measured by self-reported sleep-disturbance, which is defined as remembering events of awakening from sleep and the disturbed process of falling asleep. If individuals define their sleep as “very” or “extremely” disturbed they are classified as highly sleep disturbed [1].

Cardiovascular diseases can be split into ischemic heart disease (IHD) and hypertension. Hypertension is not included in this assessment because in the most recent burden of disease studies, hypertension is not considered as a health outcome but rather as a risk factor for other diseases [23,24]. Cognitive impairment was not included in this assessment because the burden of disease resulting from such impairments cannot be quantified by DALYs. Hearing impairment and tinnitus are especially relevant for leisure type situations such as attending night clubs or listening to personal devices. In these situations, people are exposed to noise levels which are generally higher than the exposure to traffic noise levels. As leisure activities are not addressed in the EU Environmental Noise Directive (2002/49/EC), they are not included in this assessment [2,25].

The evidence on the association between health outcomes and traffic noise exposure, which was assessed by the authors of the cited articles, varies from very low to moderate quality (see Table 1). None of the associations was rated as “high quality”. Overall the quality of evidence for road traffic noise is better rated than those for the other noise sources.

Table 2 displays the key parameters of exposure-response functions presented in the reviews and used for this assessment. For exposure-response functions for morbidity due to ischemic heart disease, only information from studies assessing the prevalence of ischemic heart disease were selected, because this calculation of the burden of disease focuses on prevalence-based estimates. For ischemic heart disease mortality attributable to railway noise no sufficient evidence is available to derive an exposure-response function [9]. All confidence intervals of the exposure-response functions except those for ischemic heart disease morbidity and road traffic noise include one. Assuming that traffic noise does not have a positive effect on ischemic heart disease values below one are set to one in the calculation. The exposure-response functions for sleep disturbance are valid from 40 to 65 decibel (dB). For people exposed to levels above 65 dB the risk for sleep disturbance of 65 dB is applied.

The counterfactual values for ischemic heart disease are chosen according to WHO recommendations in their Environmental Noise Guidelines for the European Region for the weighted average of the lowest noise levels measured in underlying studies. For road traffic noise the counterfactual value is 53 dB *L_den_* and for aircraft noise 47 dB *L_den_*. As there is no value available for railway noise 53 dB *L_den_* is used as a proxy based on the value for road traffic noise [2].

### 2.3. Estimation of the Number of Persons Exposed to Traffic Noise in Germany

The EU Environmental Noise Directive (2002/49/EC) provides information on the noise situation in Europe [25]. The aim of this directive is to reduce environmental noise and prevent noise from increasing in hitherto quiet areas. The first noise mapping took place in 2007 and has since been repeated every five years. Noise maps are developed for all major roads, major railways, and major airports as well as agglomerations (areas with more than 100,000 inhabitants). Major roads are regional, national or international roads, which have more than three million vehicles passages per year. Mayor railways are defined as railways with more than 30,000 train passages per year. A major airport is defined as a civil airport with more than 50,000 aircraft movements per year. The noise exposure of the previous calendar year has to be determined for road traffic, rail traffic, and air traffic as well as for industrial sites. To ensure comparability of the results, consistent parameters are used across the EU, namely the day-evening-night noise index *L_den_* and the night-time noise index *L_nigh_*_t_. The *L_den_* is an indicator for the entire day and contains supplements of 5 dB for the evening and 10 dB for the night-time. In Germany, the evening is defined with hours from 18 to 22 pm and the night-time from 22 to 6 pm. Additionally, the noise index *L_night_* describes the noise exposure separately at night-time. 

The regulations for calculating noise attributable to road, railway and air traffic, as well as industry, are described in detail in subordinate regulations of the German Federal Immission Control Act [26]. These regulations are interim noise calculation procedures which are based on the national calculation procedures and meet the requirements of the Environmental Noise Directive. However, an important goal of the Directive is to determine noise exposure using uniform criteria. For this reason, the European Commission introduced Common Noise Assessment Methods in the EU (CNOSSOS-EU) which have been implemented in Annex II to Environmental Noise Directive in 2015 [27]. These methods comprise detailed calculation procedures for traffic noise and noise from industry. They were transposed into German law in December 2018 and replace the so far used interim noise calculation procedures [28]. The new noise calculation procedures will be applied for the next noise mapping in 2022.

For the current noise mapping in 2017, the interim noise calculation procedures were used. The mapping was carried out for 70 agglomerations, 49,000 kilometers of major roads, 14,000 kilometers of major railway lines, and eleven major airports in Germany [21]. Table 3 shows the number of persons affected by *L_den_* levels above 55 dB for the different traffic noise sources. Furthermore, the number of people exposed to *L_night_* levels above 50 dB is indicated. The calendar year of 2016 is used as reference. 

### 2.4. Health Data and Disability Weights

Population data and data on life expectancy in Germany are obtained from the Federal Statistical Office [30,31]. Both are stratified by five-year age groups and sex. The reference year for the population data is 2016. Life expectancy refers to the period 2015–2017.

Mortality data for ischemic heart disease is obtained from the German Federal Health Monitoring [32]. This information is based on the cause-of-death statistics for Germany, which are compiled by the Federal Statistical Office. The data on deaths with the ICD 10 Codes I20–I25 were downloaded on 07/02/2019. The reference year is 2016. The data are stratified by five-year age groups and sex.

There is no official registry on morbidity data in Germany. Therefore, the number of prevalent ischemic heart disease cases were quantified by using prevalence rates estimated in the “Gesundheit in Deutschland aktuell” (GEDA) study. The rates are stratified by five age groups (18–44, 45–54, 55–64, 65–74, ≥70 years) and sex [33].

The disability weights recommended in the WHO Environmental Noise Guidelines for the European Region are used [2]. Except for ischemic heart disease, the value given by the WHO refers to acute myocardial infarction, but our assessment considers the prevalence of ischemic heart disease. Thus, applying a disability weight for the acute state would result in an overestimation of the disease burden. Another source for disability weights is the global burden of disease study, which is the most comprehensive burden of disease assessment to date [34]. However, it does not provide a disability weight for the overall group of ischemic heart disease. Ischemic heart disease is split into several acute and chronic health states and thus, at least in the global burden of disease study, the disease burden is not calculated for ischemic heart disease as a group. For the main analysis, the disability weight for angina pectoris form the global burden of disease 2017-study was used to approximate the general effects of ischemic heart disease [35]. Here, a weighted average for different severities of this health state was compiled resulting in a summary disability weight of 0.114 [36].

A compilation of the data used is shown in Table 4.

### 2.5. Sensitivity Analysis

In a sensitivity analysis, several parameters of the burden of disease quantification are altered to test the robustness of the model and whether other data may be more applicable. For this purpose, ten scenarios were considered (see Table 5 ). In the scenarios the main parameters are varied. The following input data have an influence on the results:

Exposure data: Several studies recommend to use local data to assess the number of people feeling highly annoyed by a given environmental noise source [24,37]. Every second year the German Environment Agency conducts a representative study to assess the environmental awareness and environmental behavior [38]. In the last study from 2016, the noise annoyance by traffic noise was assessed by asking about noise disturbance and annoyance in the last 12 months due to road traffic, air traffic and rail transport [22]. The percentage of highly annoyed was defined by all participants who selected ‘very’ or ‘extremely’ as answer options. Combining the two highest categories is in agreement with the WHO burden of disease recommendations for environmental noise [37]. This percentage is applied to the entire German population above 15 years of age, which is 34,968,957 males and 36,414,888 females in 2016 [30]. Only people above 15 years of age are considered because the study merely considers participants above the age of 14 years, and the statistics are grouped in five-year age groups.

Exposure-response function: Alternative exposure-response functions were used for annoyance as recommended by the European Commission [39] and sleep disturbance [37,40]. These functions were used in the WHO environmental burden of disease quantification for noise [37].

Disability weights: In the WHO environmental burden of disease publication concerning environmental noise an upper and a lower bound estimate of the disability weight is reported [37]. These bounds are used to quantify an uncertainty range of the YLDs. Ischemic heart disease also includes stroke. It is assumed that stroke, in general, leads to more severe long-term disability. Thus, a combined disability weight for stroke was estimated at 0.266. This is a weighted average over the different stroke disability weights from the global burden of disease 2017-study [35,41]. Van Kamp et al. [24] recommend a disability weight for sleep disturbance of 0.0175 and for annoyance a disability weight of 0.01. This is a guidance document for performing an environmental noise and health assessment for local authorities.

Counterfactual value: For the counterfactual value van Kamp et al. recommend to use 53 dB for a health risk assessment for all traffic noise sources [24].

## 3. Results

### 3.1. Ischemic Heart Disease

The YLLs, YLDs, and the sum of both, the DALYs, for ischemic heart disease are quantified, for each age group and both sexes (see Figure 2). The absolute numbers reveal an overall trend of the burden increasing with age. This is related to the increased number of deaths and cases of illness in higher age groups. For YLLs the burden for males is highest in the age group 75 to 79 years whereas for females the highest total burden is observed in the age group 85 to 89 years and is much smaller compared to the male burden when considering only absolute numbers (131,260 YLLs in males, 92,394 YLLs in females). A similar trend is observed for the YLDs with males having a higher burden than females. Only in the highest age groups, the burden for females is higher compared to the one for males. The highest number of YLDs for both sexes is in the age group 75 to 79 years (52,480 YLDs in males, 43,731 YLDs in females). YLDs contribute to around 25% to the overall DALYs. The ratio ranges from 90% for the age group 20 to 24 years to 12% in the highest age group. 

In the next step, the share of the YLLs, YLDs and DALYs which can be attributable to traffic noise is quantified (see Table 6). For ischemic heart disease the PAF strongly varies by noise sources from 0% (lower bound of the confidence interval is below 1, e.g., ischemic heart disease mortality) to 35% (upper confidence interval for railway noise). The total burden is highest for road traffic noise with 122,800 (95% CI: 25,954–242,068) DALYs. Followed by aircraft noise with 89,236 (95% CI: 0–246,429) DALYs and railway noise with 45,987 (95% CI: 0–141,591) DALYs. However, the burden due to railway noise only comprises YLDs because no valid exposure-response function for mortality was available. The DALYs per 100,000 people show similar results concerning the burden due to the three sources.

### 3.2. Annoyance and Sleep Disturbance

The percentage of people feeling highly annoyed due to noise differs between the various noise sources. Most people feel annoyed by aircraft noise, ranging from around 30% (55–60 dB) to 60% (70–75 dB) (see Table 7). However, the total burden is lower compared to road traffic and railway noise because fewer people are exposed to aircraft noise. The highest absolute burden is due to road traffic noise with 29,433 YLDs, followed by railway noise with 23,367 YLDs. Considering the relative burden per 100,000 people being exposed to the specific traffic noise source, the highest burden is estimated for aircraft noise (670 YLDs per 100,000). The other two noise sources contribute around half of the relative burden (349 DALYs per 100,000 for road traffic noise and 364 for railway noise) compared to aircraft noise.

The total burden due to sleep disturbance and traffic noise is the highest for railway noise because numerous people are exposed to railway noise at night. This is especially the case along the freight corridors which are most busy during night times. An example of this is the Middle Rhine Valley, which is part of the European railway freight corridor between Rotterdam and Genoa. The percentage of people being sleep disturbed rises strongly with an increase in noise levels (dB) from around 8% to 25% (see Table 8). Railway noise exposure causes a burden of 39,548 YLDs related to sleep disturbance. The burden of road traffic noise concerning sleep disturbance is lower with 24,654 YLDs and aircraft noise related YLDs are much lower at 3906 YLDs. However, regarding the YLD-rate per 100,000 people exposed to noise at night, the burden is lower for railway noise (767 YLDs per 100,000) compared to aircraft noise related to sleep disturbance (1607 YLDs per 100,000).

### 3.3. Combined Burden of the Different Health Outcomes

Considering all health outcomes, the total burden due to the different traffic noise sources is 66,884 DALYs for road traffic noise, 10,166 DALYs for aircraft noise, and 62,915 DALYs for railway noise (see Table 9). However, taking into consideration the evidence level of the risk-outcome-pairs, the burden of road traffic noise is much lower when restricting the health outcomes to those showing moderate evidence (29,999 DALYs). The burden due to aircraft noise exposure is estimated at 9575 DALYs when considering only moderate evidence because the association of aircraft noise and ischemic heart disease is valued with very low (morbidity) and low (mortality) evidence.

### 3.4. Sensitivity Analyses

Table 10 shows the burden of traffic noise related ischemic heart disease as a result of changing the counterfactual value and the disability weights. Considering a higher disability weight of 0.266 results in a doubling of the burden compared to the main analysis. The change of the counterfactual value only impacts the results for aircraft noise, where the counterfactual value was increased from 47 dB to 53 dB. This leads to a lower burden reduced by around a half from 89,236 (95% CI: 0–246,429) in the main analysis to 43,829 (95% CI: 0–124,342) DALYs in scenario ten.

Using alternative data for the health outcome annoyance allows quantifying the burden for females and males separately (see Table 11). However, the estimates show no relevant differences between males and females (scenario 1). Comparing the three noise sources, most people report feeling highly annoyed by road traffic noise, with 23% for both sexes. Followed by aircraft noise, 10% of males and 8% of females and railway noise 7% of males and 6% of females. According to this the burden quantified in YLDs is the highest for road traffic noise with 328,366 YLDs, which is 10 times higher compared to the main assessment. For aircraft noise, the burden is 128,202 YLDs, which is even 20 times higher. For railway noise, the burden in the alternative scenario with 92,654 YLDs is four times higher. Applying the exposure-response function recommended by the European Commission [39] results in a decrease in the burden. For aircraft noise, the analysis shows the strongest decline of around half the burden compared to the main analysis with 132,352 YLDs. For railway noise around a third of the burden compared to the main analysis with 813,337 YLDs were calculated. Applying the lower and upper bounds of the disability weights allows to estimate a range around the YLDs, which can be interpreted as an uncertainty interval around the estimates quantified in the main analysis. Using the disability weight of 0.0175 in scenario seven, as recommended by van Kamp et al. [24] leads to a slightly lower burden.

In the sensitivity analysis for sleep disturbance, a different exposure-response function and alternative disability weights were applied (see Table 12). Applying the exposure-response function, as proposed by Miedema [40], results in a higher number of YLDs due to road traffic noise (101,371 more YLDs compared to the main analysis) and less YLDs due to aircraft noise (33,872 less YLDs compared to the main analysis) and railway noise (355,843 less YLDs compared to the main analysis). Scenarios four and five are based on the lower and upper limits of the disability weight used in the main analysis. 

## 4. Discussion

Considering all of the included health outcomes regardless of the evidence of their association, the results indicate that the total burden of disease is 176,888 DALYs for road traffic noise, 98,810 DALYs for aircraft noise and 108,902 DALYs for railway noise. As the assessment of the burden of disease relies on a number of assumptions, it involves possible uncertainties, as well as limitations and restrictions. In the following, the main limitations of the current assessment are discussed and the results are compared to other studies.

### 4.1. Over- and Underestimation

The burden of disease attributable to traffic noise was estimated at a total of 348,600 DALYs when considering the three traffic noise sources and the three health outcomes regardless of the evidence of their association. There are several hints that this sum can be an over- as well as an underestimation. One reason for an overestimation is that in the sum the entire burden related to the single noise sources is added up. It does not consider people exposed to more than one noise source simultaneously [24]. Therefore, the impact of exposure to multiple noise sources is not taken into account [42]. According to the German representative study about environmental awareness and environmental behavior more than the half of the surveyed people are exposed to more than one noise source: Around 28% of the subjects state that they feel annoyed due to two sources and around 25% feel annoyed because of three sources [38]. The joined assessment of the health effects is still under research [43]. It is further still unclear whether the different effects are additive [37]. 

Another factor which could lead to an overestimation is a possible overlap between different outcomes, where, e.g., a person who feels annoyed by noise also might have or develop ischemic heart disease. The cumulative effects of the exposure to various noise sources at the same time are complex. There is an ongoing scientific discussion about the best way to model combined noise exposure and how to assess the combined health effects. Both models of energetic summation and effect-related weighting have been developed [44,45,46,47]. The model of energetic summation postulates an energy equivalent summation of the average sound pressure levels of all single noise sources. The model assumes that each individual noise source has the same impact on a given health effect (e.g., annoyance). Effect-related models, on the contrary, incorporate effect-related variance of singular noise sources into a weighted equation according to the premises of the model. There is a multitude of different effect-related approaches [48], but in general, there is a growing body of evidence on the relevance of the consideration of effect-related variance for the estimation of total noise exposure [47,49,50]. Similarly, there is still an ongoing scientific discourse about the joined assessment of the adverse health effects due to multiple noise exposures.

Another factor which leads to an overestimation is the application of the remaining life expectancy above 90 years to all deaths above 90 years (22% of deaths occur in the age group above 90 years). However, people dying at aged 100 years would statistically have a marginally lower remaining life expectancy than the people dying at the age of 90 years.

One reason why it could be an underestimation is that the actual number of people affected by noise in Germany is higher because the EU Environmental Noise Directive only considers major noise sources. For instance, only the eleven busiest German airports fall under the Directive. Smaller commercial airports and military airfields are not covered by the Directive although they produce considerable noise levels. The number of people affected by noise in Germany outside the scope of the Environmental Noise Directive is unknown because there are no reliable data available. 

The exposure-response function for sleep disturbance is only applicable up to a noise level of 65 dB. However, around 320,700 people in Germany are exposed to night noise levels above 65 dB. In the burden of disease quantification for these people, the risk for sleep disturbance at 65 dB was applied, which is most probably an underestimation.

Moreover, the environmental burden of disease assessments relies on exposure-response functions. Thus, health outcomes can only be included in an assessment if an exposure-response function is available. For traffic noise exposure, the number of outcomes meeting this criterion is limited [2]. However, studies show that there are more health effects which might be associated with environmental noise, such as cognitive impairments, learning difficulties, psychological disorders like depressive episodes and other chronic diseases as diabetes or obesity [3,4,5,9,51,52,53]. With growing evidence in this field, considering these outcomes in a health risk assessment would lead to considerably higher estimates of the disease burden due to traffic noise. 

### 4.2. Comparison to Other Studies

The biggest share of the total burden attributable to traffic noise is due to ischemic heart disease. Ischemic heart diseases contribute 90% to the DALYs related to aircraft noise. For road traffic and railway noise, the share is lower with 69% and 42%, respectively. These results differ considerably from WHO estimates for western European countries. In the WHO assessment, sleep disturbance contributes the most to the DALYs with a share of more than 50%, followed by annoyance with around 40% and ischemic heart disease about 4%. Whereas ischemic heart disease burden was only quantified for road traffic noise the burden due to sleep disturbance and annoyance was quantified for all three traffic noise sources. The difference in the share of the health outcomes in the total burden can be explained due to differences in the used exposure-response functions. The PAF estimated by the WHO is 1.8% compared to the PAFs in this investigation ranging from 4% to 18% [54]. Thus, the currently recommended exposure-response functions in the WHO Environmental Noise Guidelines for the European Region result in a higher attributable burden for ischemic heart disease.

In particular, for Germany, the burden of disease related to traffic noise was quantified in two assessments [55,56]. In the project “Environmental Burden of Disease in European countries” (EBoDE) sleep disturbance accounts for 94% of the DALYs [56]. This project has considered only sleep disturbance and ischemic heart disease attributable to road traffic noise. In total 48,770 DALYs were quantified, from which 94% (45,844 DALYs) are due to sleep disturbance. With around 68,000 DALYs the burden related to sleep disturbance due to the three noise sources quantified in this study is much higher. The reasons for the difference are the updated exposure data (in the EBoDE project, the noise exposure data from the previous noise mapping 2012 were used) and the updated exposure-response functions. When considering the Miedema curves from 2004 (as used in EBoDE) the burden estimate with around 50,000 DALYs in scenario two of this sensitivity analysis is closer to results from EBoDE [56]. 

In the German project “Distribution-based analysis of the health effect of environmental stressors” (VegAS) the burden attributable to traffic noise is quantified for different health outcomes [55]. The burden of disease estimates for the three noise sources include the health outcomes annoyance and sleep disturbance. For road traffic noise also the burden due to hypertension, stroke and myocardial infarction was quantified. In the VegAS-study annoyance accounts for 18,843 DALYs, which is less than a third of the DALYs quantified in our assessment for traffic noise and annoyance (58,469 DALYs). Likewise, the burden for sleep disturbance is higher in our assessment than in the VegAS-study. Even when applying the same exposure-response function used in the VegAS-study for annoyance the burden estimated in our study is much higher (31,612 DALYs) compared to VegAS (18,843 DALYs) [55]. Both studies (EBoDE and VegAS) show a lower burden, which is due to less considered health outcomes, outdated exposure-response functions, exposure data from 2012, and health statistics from previous years. 

In line with the results of various studies and reviews about the impact of traffic noise on noise annoyance, the present study confirms the need for a modification of the exposure-response-functions that are currently applied in the EU Environmental Noise Directive [25]. People feel more annoyed by traffic noise (especially railway and aircraft noise) at the same average sound pressure level nowadays, than they have been twenty years ago [8,52]. This increase in annoyance at the same average sound pressure level is reflected in the new WHO Environmental Noise Guidelines for the European Region. The percentage of 10% highly annoyed individuals due to railways noise and aircraft noise is reached about 10 dB earlier according to the exposure-response function in the WHO Environmental Noise Guidelines for the European Region in comparison to the exposure-response function by Miedema, which is currently applied in the Environmental Noise Directive [2,25]. The increase in annoyance is especially prominent for railway noise and aircraft noise. In comparison, the increase in percentage feeling annoyed by road traffic noise is less intense for lower exposure levels and almost remains stable at higher exposure levels [2]. The European Commission has recognized the need for modification and is currently revising Annex III of the EU Environmental Noise Directive. Annex III defines exposure-response functions for various health outcomes and environmental noise. 

### 4.3. Evidence Rating and Influence on the Results

The evidence rating according to Grading of Recommendations Assessment, Development and Evaluation (GRADE) of ischemic heart disease and noise is diverse (see Table 1), ranging from not existing for railway noise and ischemic heart disease mortality to moderate evidence for road traffic noise and ischemic heart disease mortality [1,8,9]. Yet, except for road traffic noise and ischemic heart disease mortality the evidence for all ischemic heart disease relationships is considered to be very low or low quality. Therefore, the results for ischemic heart disease should be interpreted with caution. Nevertheless, there is a strong biological plausibility for the negative effects of environmental noise due to aircraft noise and road traffic noise on the cardiovascular system [5,24]. Furthermore, more recently published studies not yet considered in the evidence reviews by the WHO, also feed into the pool of growing evidence on the association of environmental noise and cardiovascular diseases [57,58]. 

Regarding annoyance and sleep disturbance, the evidence is rated as moderate quality except for annoyance and road traffic with is rated as low quality. Nevertheless, the evidence is rated higher than for ischemic heart disease. The burden due to ischemic heart disease is much higher compared to both other health outcomes. Taking into account only moderate evidence the burden is much lower with a total of 148,386 DALYs, which is mainly due to sleep disturbance (68,109 YLDs) and annoyance (29,035 YLDs). 236,215 DALYs are estimated based on a low evidence association between traffic noise and health outcomes (mainly ischemic heart disease with 206,781 DALYs).

### 4.4. Input Data

In Germany, no comprehensive register exists for the prevalence of diseases. Therefore, data from a study were used to estimate the number of ischemic heart disease cases. Prevalence rates were collected in the “Gesundheit in Deutschland aktuell” (GEDA)-study [33]. The rates are stratified by five age groups (18–44, 45–54, 55–64, 65–74, ≥70 years) and sex. It is a self-reported question which could be answered by telephone or online. It refers to the presence of myocardial infarction, chronic consequences of myocardial infarction, coronary heart disease or angina pectoris in the last 12 months. The data are not validated by medical professionals and thus include some uncertainty.

The counterfactual value for ischemic heart disease is arbitrary, because there is no scientific evidence on a threshold for the occurrence of health effects due to traffic noise. In this case, several options are possible for assessing the burden of disease. Most common is the use of the lowest level measured in the epidemiological studies, which are taken as input data for the exposure-response function. In the systematic reviews for ischemic heart disease a range of exposure for the relative risk is given, e.g., under table A28 page 31: “We found a non-significant effect size of 1.04 per 10 dB across a noise range of 40 to 60 dB”. As the effect-size was not significant and applying 40 dB would result in a very high burden, the weighted average of the lowest noise levels measured in underlying studies included in the WHO Environmental Noise Guidelines for the European Region was used. For railway noise, the WHO does not indicate any exposure limits for ischemic heart disease, due to a lack of scientific evidence. Thus, the road traffic value was used as a proxy for railway noise. Acknowledging that this transfer can be uncertain and the input data has a strong effect on the results a second approximation was additionally included. Van Kamp et al. [21] recommend a counterfactual value of 53 dB, which was used in the sensitivity analysis, resulting in a lower burden for aircraft noise, because in the main analysis the counterfactual value was 47 dB (45,407 less DALYs).

To quantify the burden of disease the prevalence approach was used, as this approach does not rely on the assessment of the duration of disease. The duration of a disease is very diverse and depends on various factors. In Germany, no reliable information for the duration of ischemic heart disease exists. The WHO recommends relative risks for incidence, but for the prevalence approach, specific relative risks for prevalence are needed. These data were available in the systematic reviews, which are used for the preparation of the WHO Environmental Noise Guidelines for the European Region. The used approach also differs from the one recommended by van Kamp et al. [24]. They also go for the incidence approach, thus different relative risks were used. Likewise, they recommend using one relative risk for all noise sources. For mortality due to ischemic heart disease, a relative risk of 1.05 is recommended. This would provide higher estimates compared to our assessment because for aircraft noise a relative risk of 1.04 was used and for railway noise no relative risk was available and thus no burden quantified.

The Environmental Noise Directive provides comparable and broad-based modeled data for noise exposure in Europe. Consistent indices (*L_den_* and *L_night_*) and adapted calculation methods are used in all EU Member States. Exact thresholds for the considered transport infrastructure (roads, railways, airports) are binding and improve the quality of the data. Thus, a reliable data basis for calculating the environmental burden of disease attributable to noise is already in place. The Common Noise Assessment Methods in the EU (CNOSSOS-EU) [27] which will have to be used starting with the next round of noise mapping in 2022 will further improve the quality and comparability of noise exposure data between the Member States. Nevertheless, there is still potential for further improvement in comparability due to interpretation possibilities in the application of the methods.

Outside agglomerations there are exact traffic regulations for noise mapping. Inside agglomerations, different qualities of input data are used by the competent authorities. Some agglomerations consider major roads and the main road network. Others mapped only roads down to a certain threshold, for example, 1.5 million vehicles per year. Therefore, exposure data for different agglomerations are not directly comparable. There is also room for improvement outside the Environmental Noise Directive. For example, this Directive only covers major roads outside agglomerations with more than three million vehicles per year. Noise exposure from all roads together, however, is higher. Smaller roads with less than three million vehicles also can have a relevant impact on noise exposure. A report by the European Topic Centre on Air Pollution and Climate Change Mitigations shows that total exposure from road noise is more than twice as high as covered by the Environmental Noise Directive [59]. To close this gap between Environmental Noise Directive and total exposure in Germany the German Environment Agency is funding a research project. The results are expected in 2020. With these results, more detailed models to calculate the environmental burden of disease attributable to noise are possible.

The Environmental Noise Directive data start at 55 dB at daytime and 50 dB at night, which is not enough for a comprehensive burden of disease assessment [24]. By assuming that the risk of ischemic heart disease is one at 53 dB for road traffic, the people exposed to dB levels below 53 dB are regarded to be without any burden. The counterfactual value for the exposure to aircraft noise is 47 dB, and thus for a complete burden assessment people exposed to this low level would be needed. Furthermore, we do not know from empirical studies how people are distributed within the 5-dB classes. Therefore, we assume an equal distribution of people in 5-dB classes of exposure.

Using the exposure data for noise annoyance based on the environmental awareness and environmental behavior study of the Germany population leads to a much higher burden compared to the recommended exposure-response functions of the WHO. Further research is needed to assess if noise annoyance is region specific.

In all relevant scenario analyses, disability weights appeared to be a sensitive modeling parameter. Altering the disability weight had a strong impact on the results, e.g., showing a doubling of the disease burden for ischemic heart diseases when considering the higher disability weight of 0.266. Thus, the choice of an adequate disability weight remains a crucial factor. In most cases of our main analyses, we used a conservative approach to avoid overestimation of disease burden. 

Noise and air pollutants often share their sources, however, the correlation between these pollutants and their variability remains uncertain. The correlations are clearly dependent on the single pollutant and its source. Thus, mutual confounding is often discussed in studies which cover either noise or air pollution. Though, there are also studies indicating the independent effects of noise even after adjustment for air pollutants. Héritier et al. [60] investigated the association between noise and myocardial infarction and found stable estimates even when adjusted for major pollutants such as particulate matter or nitrogen dioxide. They rather showed that effect estimates for air pollutants decreased after noise adjustment indicating an overestimation of the air pollution effect when estimates are not adjusted for noise exposure.

The burden of estimate includes the following assumptions:

### 4.5. Necessary Assumptions to Quantify the Burden of Disease:

People in five-decibel classes are equally distributed.The exposure-response functions are applicable for the population in GermanyThe prevalence rates assessed in the GEDA-study can be applied to entire population in GermanyThe mortality data are validBelow the counterfactual value, the risk of ischemic heart disease is one

## 5. Conclusions

In the global burden of disease-study, traffic noise is not included as a risk factor, which could lead to the assumption that traffic noise does not cause health losses in terms of DALYs. However, we could show that traffic noise is a harmful risk factor which causes a considerable burden. The burden of disease assessment for noise is becoming more sophisticated and reliable, with the increased availability of input data, e.g., relative risks recommended by the WHO and an improving evidence base on the association between noise and health outcomes from epidemiological studies. Therefore, the number of DALYs which can be attributed to traffic noise is higher than in the previous studies. The noise mapping according to the EU Environmental Noise Directive is the most suitable source for exposure data. Nevertheless, due to uncertainties in the input data, the results should still be interpreted with caution. 

## Figures and Tables

**Figure 1 ijerph-16-02304-f001:**
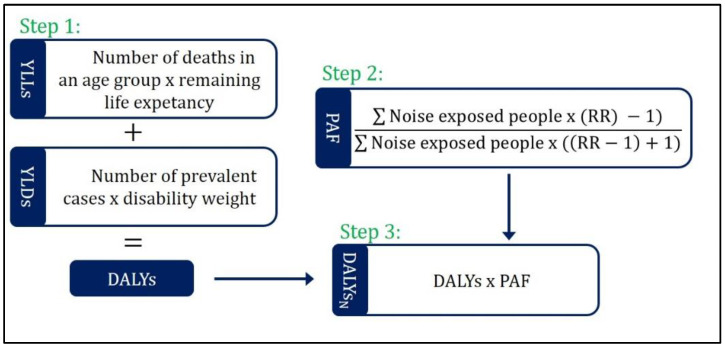
Technical roadmap of the burden of disease quantification referring to [20,21], YLLs = years of life lost due to premature mortality, YLDs = years lived with disability, DALYs = disability-adjusted life years, PAF = population attributable fraction, RR = relative risk, DALYs_N_ = DALYs attributable to noise exposure.

**Figure 2 ijerph-16-02304-f002:**
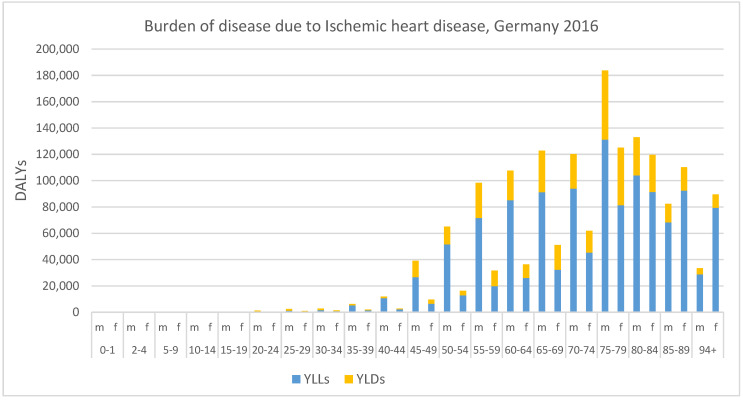
Burden of disease due to ischemic heart disease in Germany 2016, expressed in total YLLs, YLDs and DALYs for males (m) and females (f) by age groups, YLLs = years of life lost due to premature mortality, YLDs = years lived with disability, DALYs = disability-adjusted life years.

**Table 1 ijerph-16-02304-t001:** Evidence on the association between traffic noise exposure and health outcomes.

Health Outcomes	Evidence Quality Assessment for			Source
	Road Traffic Noise	Aircraft Noise	Railway Noise	
IHD morbidity	••	•	•	[9]
IHD mortality	•••	••	*	[9]
Annoyance	••	•••	•••	[8]
Sleep disturbance	•••	•••	•••	[1]

IHD = ischemic heart disease, • = very low quality, •• = low quality, ••• = moderate quality, * no studies available.

**Table 2 ijerph-16-02304-t002:** Exposure-response functions for road, aircraft and railway noise.

Health Outcomes	Exposure-Response Functions for			Source
	Road Traffic Noise	Aircraft Noise	Railway Noise	
IHD morbidity	1.24 (95% CI: 1.08–1.42) **	1.07 (95% CI: 0.94–1.23) **	1.18 (95% CI: 0.82–1.68) **	[9]
IHD mortality	1.05 (95% CI: 0.97–1.13) **	1.04 (95% CI: 0.98–1.11) **	-*	[9]
Annoyance	78.9270 − 3.1162 × *L_den_ *+ 0.034 × *L_den_*^2^	−50.9693 + 1.0168 × *L_den_ *+ 0.0072 × *L_den_*^2^	38.1596 − 2.05538 × *L_den_ *+ 0.0285 × *L_den_*^2^	[8]
Sleep disturbance	19.4312 − 0.9336 × *L_night_ *+ 0.0126 × *L_night_*^2^	16.7885 − 0.9293 × *L_night_ *+ 0.0198 × *L_night_*^2^	67.5406 − 3.1852 × *L_night_* + 0.0391 × *L_night_* ^2^	[1]

IHD = ischemic heart disease, CI = confidence interval, * no studies available, ** per 10 dB linear increase.

**Table 3 ijerph-16-02304-t003:** Number of persons affected by different traffic noise sources in Germany in 2016.

*L*_den_/*L*_night_ in dB	Number of People Exposed to					
	Road Traffic Noise		Aircraft Noise		Railway Noise	
	*L* _den_	*L* _night_	*L* _den_	*L* _night_	*L* _den_	*L* _night_
>50–55		2,835,300		206,800		3,144,400
>55–60	3,822,500	1,772,300	606,400	34,700	3,782,700	1,310,700
>60–65	2,349,000	726,400	205,800	1500	1,639,300	483,600
>65–70	1,603,200	97,000	30,700		679,900	154,700
>70–75	600,300	6800 *	3700		230,700	62,200 *
>75	60,100				92,500	

Source: [29], * people exposed to *L_night_* above 70 dB.

**Table 4 ijerph-16-02304-t004:** Input data used for the calculation of the burden due to traffic noise exposure in Germany.

Health Outcomes	Health Data	Reference Year	Source	DW	Source
IHD morbidity	GEDA-study	2014/2015	[33]	0.114	*
IHD mortality	Cause of death register	2016	[32]	-	-
Annoyance	-			0.02	[2]
Sleep disturbance	-			0.07	[2]

IHD = ischemic heart disease, DW = disability weight, GEDA = Gesundheit in Deutschland aktuell, * own quantification based on [35,36].

**Table 5 ijerph-16-02304-t005:** Alternative input data used in the sensitivity analysis.

Scenario	Road Traffic Noise	Aircraft Noise	Railway Noise	Source
S 1: Exposure data	%HA (23% males, 23% females)	%HA (10% males, 8% females)	%HA (7% males, 6% females)	[38]
S 2–3: Exposure-response function	%SD (20.8 − 1.05 × (*L_night_*) + 0.01486 × (*L_night_*)^2^) %HA (9.868 × 10^−4^ × (*L_den_*-42)^3^ − 1.436 × 10^−2^ × (*L_den_*-42)^2^ + 0.5118(*L_den_*-42))	%SD (18.147 − 0.956 × (*L_night_*) + 0.01482 × (*L_night_*)^2^) %HA (−9.199 × 10^−5^ × (*L_den_*-42)^3^ + 3.932 × 10^−2^ × (*L_den_*-42)^2^ + 0.2939 (*L_den_*-42))	%SD (11.3 − 0.55 × (*L_night_*) + 0.00759 × *L_night_*)^2^) %HA (7.239 × 10^−4^ × (*L_den_*-42)^3^ − 7.851 × 10^−3^ × (*L_den_*-42)^2^ + 0.1695 (*L_den_*-42))	[37,39,40]
S 4–9: Disability weight	IHD (0.266) %HA (0.01–0.12, 0.01) %SD (0.04–0.10, 0.0175)			[24,35,37,41]
S 10: Counterfactual value for IHD	53 dB *L_den_*	53 dB *L_den_*	53 dB *L_den_*	[24]

IHD = ischemic heart disease, S = scenario, %HA = % highly annoyed, %SD = % sleep disturbed.

**Table 6 ijerph-16-02304-t006:** Burden of disease due to ischemic heart disease attributable to traffic noise in Germany, 2016 (95% CI in brackets).

Health Outcomes	Road Traffic Noise		Aircraft Noise		Railway Noise	
	PAF	Life Years Lost	PAF	Life Years Lost	PAF	Life Years Lost
Morbidity	1.83% (0.65–2.93%)	7452 (2706–11,905) YLDs	0.08% (0.00–0.24%)	327 (0–976) YLDs	0.92% (0.00–2.8%)	3738 (0–11,402) YLDs
Mortality	0.42% (0.00–1.05%)	5345 (0–13,299) YLLs	0.05% (0.00–0.12%)	591 (0–1557) YLLs	–	–
Sum		12.797 (2706–25,204) DALYs		918 (0–2533) DALYs		3.738 (0–11.402) DALYs
Per 100,000 *		15.54 (3.29–30.61) DALYs		1.11 (0–3.08) DALYs		4.54 (0–13.85) DALYs

PAF = population attributable fraction, YLLs = years of life lost due to premature mortality, YLDs = years lived with disability, DALYs = disability-adjusted life years, IHD = ischemic heart disease, CI = confidence interval, * people exposed to the related traffic noise source.

**Table 7 ijerph-16-02304-t007:** Burden of disease due to annoyance attributable to traffic noise in Germany, 2016.

L_den_ in dB	Road Traffic Noise			Aircraft Noise			Railway Noise		
	in %	#	YLDs	in %	#	YLDs	in %	#	YLDs
>55–60	12.4	474,732	9495	30.4	184,231	3685	13.6	514,426	10,289
>60–65	17.2	403,732	8075	39.8	81,804	1636	20.3	332,451	6649
>65–70	23.7	379,404	7588	49.5	15,189	304	28.4	192,994	3860
>70–75	31.9	191,216	3824	59.6	2204	44	37.9	87,473	1749
>75	37.6	22,590	452	65.8	0	0	44.3	40,995	820
Sum		1,471,673	29,433		283,236	5669		1,168,338	23,367
Per 100,000 *			349			670			364

YLDs = years lived with disability, # = number of people feeling highly annoyed, * people exposed to the related traffic noise source.

**Table 8 ijerph-16-02304-t008:** Burden of disease due to sleep disturbance attributable to traffic noise in Germany, 2016.

L_night_ in dB	Road Traffic Noise			Aircraft Noise			Railway Noise		
	in %	#	YLDs	in %	#	YLDs	in %	#	YLDs
>50–55	5.0	140,472	9833	22.0	45,504	3185	7.6	240,125	16,809
>55–60	7.2	126,780	8875	28.2	9768	684	13.0	170,654	11,946
>60–65	10.0	72,512	5076	35.3	529	37	20.3	98,454	6892
>65–70	12.0	11,623	814	40.0	0	0	25.7	39,758	2783
>70	12.0	815	57	40.0	0	0	25.7	15,985	1119
Sum		352,201	24,654		55,801	3906		564,977	39,548
Per 100,000 *			454			1607			767

YLDs = years lived with disability, # = number of people feeling sleep disturbed, * people exposed to the related traffic noise source.

**Table 9 ijerph-16-02304-t009:** Sum of the DALYs attributable to traffic noise by evidence quality.

Health Outcomes	Road Traffic Noise	Aircraft Noise	Railway Noise
IHD morbidity	7452	327	3.738
Evidence	••	•	•
IHD mortality	5345	591	*
Evidence	•••	••	*
Annoyance	29,433	5669	23,367
Evidence	••	•••	•••
Sleep disturbance	24,654	3906	39,548
Evidence	•••	•••	•••
Sum (moderate evidence)	29,999	9575	62,915
Sum (low quality and better)	66,884	10,166	62,915
Sum (all)	66,884	10,493	66,653

IHD = ischemic heart disease, • = very low quality, •• = low quality, ••• = moderate quality, * no studies available.

**Table 10 ijerph-16-02304-t010:** Sensitivity analysis of the burden of disease due to ischemic heart disease attributable to traffic noise in Germany, 2016 (95% CI in brackets).

Scenarios	Road Traffic Noise	Aircraft Noise	Railway Noise
Main analysis YLDs	71,558	31,773	45,987
	(25,954–114,473) YLDs	(0–94,953) YLDs	(0–141,591) YLDs
Main analysis YLLs	51,242	57,462	-
	(0–127,595) YLLs	(0–151,475) YLLs	
Main analysis DALYs	122,800	89,236	45,987
	(25,954–242,068) DALYs	(0–246,429) DALYs	(0–141,591) DALYs
S4: DW	166,969	74,138	107,303
	(60,559–267,104) YLDs	(0–221,558) YLDs	(0–330,380) YLDs
Difference in YLDs	+95,411	+42,365	+61,316
	(34,605–152,631) YLDs	(0–126,605) YLDs	(0–188.789) YLDs
S10: counterfactual value 53 dB	122,800	43,829	45,987
	(25,954–242,068) DALYs	(0–124,342) DALYs	(0–141,591) DALYs
Difference in DALYs	0	−45,407	0
		(0–122,087) DALYs	

YLLs = years of life lost due to premature mortality, YLDs = years lived with disability, DALYs = disability-adjusted life years, CI = confidence interval, DW = disability weight, S = scenario.

**Table 11 ijerph-16-02304-t011:** Sensitivity analysis of the burden of disease due to annoyance attributable to traffic noise in Germany, 2016.

Scenarios	Road Traffic Noise			Aircraft Noise			Railway Noise		
	in %	#	YLDs	in %	#	YLDs	in %	#	YLDs
Main analysis	17	1,471,673	29,433	33	283,428	5669	18	1,168,338	23,367
S1a: exposure m	23	8,042,860	160,857	10	3,496,896	69,938	7	2,447,827	48,957
S1b: exposure f	23	8,375,424	167,508	8	2,913,191	58,264	6	2,184,893	43,698
S1: exposure sum		16,418,284	328,366		6,410,087	128,202		4,632,720	92,654
Difference		+14,949,611	+298,932		+6126,659	+122,533		+3,464,382	+69,288
S3: ERF	13	1,093,241	21,865	16	132,352	2647	6	355,001	7100
Difference		−378,432	−7569		−151,076	−3022		−813,337	−16,267
S5: DW low			14,717			2834			11,683
Difference			−14,717			−2834			−11,683
S6: DW high			176,601			34,011			140,201
Difference			+147,167			+28,343			+116,834

YLDs = years lived with disability, # = number of people feeling highly annoyed, S = scenario, ERF = exposure-response function, DW = disability weight.

**Table 12 ijerph-16-02304-t012:** Sensitivity analysis of the burden of disease due to sleep disturbance attributable to traffic noise in Germany, 2016.

Scenarios	Road Traffic Noise			Aircraft Noise			Railway Noise		
	in %	#	YLDs	in %	#	YLDs	in %	#	YLDs
Main analysis	6	352,201	24,654	23	55,801	3906	11	564,977	39,548
S2: ERF	8	453,573	31,750	9	21,929	1535	4	209,134	14,639
Difference		+101,371	+7096		−33,872	−2371		−355,843	−24,909
S7: DW low			14,088			2232			22,599
Difference			−10,566			−1674			−16,949
S8: DW high			35,220			5580			56,498
Difference			+10,566			+1674			+16,949
S9: DW alternative			6164			977			9887
Difference			−18,490			−2929			−29,661

YLDs = years lived with disability, # = number of people feeling sleep disturbed, S = scenario, ERF = exposure-response function, DW = disability weight.

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
