# Peer review of "Burden of Disease Due to Traffic Noise in Germany"

_ijerph, 2019, doi:10.3390/ijerph16132304_

Round 1

Reviewer 1 Report

The topic of the paper is very general, and to some extent I fail to see what its actual novelty and/or contribution to knowledge is, since its conclusions are far from conclusive and the manuscript does not really provide ultimate answers to the research questions it poses.

The Introduction is very weak; there is no proper literature review. The paper only mentions the few papers in the IJERPH special issue it will be dealing with.

My understanding is that the paper aims to be a meta-review to some extent, but the methodology for developing the work does not look very rigorous. There is no clear protocol reported for data extraction from the papers of interest and/or for scoring their evidence (e.g. Tables 1, 9, etc.).

In spite of the attempts made in the Discussion section, the paper is not really clearly positioned or benchmarked towards other papers and projects available in literature.

The assumptions the authors report (last bullet-point list before the Conclusions section) do look as leading the data interpretation too far. For instance, the equal distribution of people in 5-dB classes of exposure is difficult to sustain in my view…

Conclusions are very limited and not very “conclusive”.

Author Response

Dear Reviewer,

Thank you very much for your valuable comments. We carefully considered and addressed them point by point:

Comment: The topic of the paper is very general, and to some extent I fail to see what its actual novelty and/or contribution to knowledge is, since its conclusions are far from conclusive and the manuscript does not really provide ultimate answers to the research questions it poses.

Answer: Thank you for this comment. The aim of our paper is stated in the end of the introduction:

“The aim of this study is to use the current body of evidence on the adverse health effects of traffic noise to quantify the burden of disease resulting from noise attributable to road, railway and air traffic in Germany.” (Line 48-50)

The answer is provided in the results section and summarized in the beginning of the discussion:

“Considering all of the included health outcomes regardless of the evidence of their association, the results indicate that the total burden of disease is 176,888 DALYs for road traffic noise, 98,810 DALYs for aircraft noise and 108,902 DALYs for railway noise.” (Line 346-348)

The novelty is the quantification of the disability-adjusted life years due to traffic noise in Germany. By this mean it is possible to compare the burden of the different noise sources. Likewise it is possible, to a certain extent, as described on the discussion section to add up the burden of the three traffic noise sources. And thus it gives a picture of the harmfulness of traffic noise. This is, to our knowledge, not done based on the current noise mapping and the latest WHO Environmental Noise Guidelines for the European Region. Furthermore we want to show, the influence of the new data on the results, which is part of the sensitivity analysis.

More generally, we are certain, that the burden of disease due to noise plays an important role for public health and especially for prevention and interventions measures aiming at the reduction of environmental noise. Burden of disease estimates frame the basis for e. g. health impact assessments which in general are important tools for policy making. Further, to our knowledge, these are the first estimates for Germany using the most recent exposure data and evidence on health outcomes resulting from noise exposure. We also do not think, that the topic is very general but certainly specific, as it focusses on one exposure component, traffic noise.

Comment: The Introduction is very weak; there is no proper literature review. The paper only mentions the few papers in the IJERPH special issue it will be dealing with.

Answer: Thank you very much for this hint, however, we think that this is a misinterpretation of the aim of our paper. It was not our aim to perform a literature review, because this was already done by the WHO and its colleagues. This is why we cite the WHO papers in the introduction. We thus are aware of the current state of knowledge, with respect to this issue. Our aim was to use this knowledge and combine it with further data to quantify summary measures of population health. We see the cited papers in the IJERPH special issue as crucial input for the models used to calculate our results. Where adequate, we added further citations. We however do not see the necessity to overload the introduction with numerous references.

Comment: My understanding is that the paper aims to be a meta-review to some extent, but the methodology for developing the work does not look very rigorous. There is no clear protocol reported for data extraction from the papers of interest and/or for scoring their evidence (e.g. Tables 1, 9, etc.).

Answer: We thank for this comment, but likely this again is misunderstanding corresponding with the previous comment. According to our research question, it is very clear, that we did neither aim to perform a systematic review, nor a meta-analysis. We use existing data, combine it with the environmental burden of disease approach and quantify disability-adjusted life years due to traffic noise in Germany. The method is described in the paragraph quantification method. In this paragraph we also described the data sources and why we used specific data for our burden of disease quantification.

The evidence for the association of traffic noise and health effects was systematically and comprehensively assessed in the papers in the IJERPH special issue. Therefore we did not see a need to do this again. To specify that we added the following:

“The evidence on the association between health outcomes and traffic noise exposure, which was assessed by the authors of the cited articles, varies from very low to moderate quality (Table 1)” (Line 111-112)

Comment: In spite of the attempts made in the Discussion section, the paper is not really clearly positioned or benchmarked towards other papers and projects available in literature.

Answer: Thank you for pointing on this issue. However, we see, that we embedded our results quite thoroughly into the existing body of evidence. As stated in the discussion, to our knowledge there are only two other studies quantifying the burden of disease due to traffic noise in particular for Germany. We compared our results to these studies in a specific paragraph (“comparison to other studies”, Line 399). In the entire paper we rely on the work of the World Health Organization and their report on environmental burden of disease of environmental noise and the publication “Study on methodology to perform an environmental noise and health assessment - a guidance document for local authorities in Europe” from Kamp and colleagues. For example:

Disability weights: In the WHO environmental burden of disease publication concerning environmental noise an upper and a lower bound estimate of the disability weight is reported [35]. These bounds are used to quantify an uncertainty range of the YLDs.“ (Line 221-223).

We certainly do discuss the results, as well as the taken assumptions.

Comment: The assumptions the authors report (last bullet-point list before the Conclusions section) do look as leading the data interpretation too far. For instance, the equal distribution of people in 5-dB classes of exposure is difficult to sustain in my view…

Answer: There are several assumptions needed to quantify the burden of disease. We aim to be transparent and thus communicate our assumptions. Regarding the equal distribution of people in 5-dB classes of exposure, we actually do not know from empirical studies how people are distributed within the 5-dB classes. However to use this data as exposure data and combine them with exposure-response functions, an assumption is necessary. It is a common approach in environmental burden of disease assessments. To make that clear we added the following:

“Necessary assumptions to quantify the burden of disease:” (Line 557)

And

“Furthermore, we do not know from empirical studies how people are distributed within the 5-dB classes. Therefore we assume an equal distribution of people in 5-dB classes of exposure.” (Line 535-537)

Comment: Conclusions are very limited and not very “conclusive”.

Answer: Thank you very much for this comment. We rewrite the conclusion:

“In the global burden of disease-study traffic noise is not included as a risk factor, which could lead to the assumption that traffic noise does not causes health losses in terms of DALYs. However we could show that Ttraffic noise is a harmful risk factor which causes a considerable burden. The burden of disease assessment for noise is becoming more sophisticated and reliable, with increased availability of input data, e. g. relative risks recommended by the WHO and an improving evidence base on the association between noise and health outcomes from epidemiologic studies. Therefore the number of DALYs which can be attributed to traffic noise is higher as in the previous studies. The noise mapping according to the EU Environmental Noise Directive is the most suitable source for exposure data. Nevertheless,. However due to uncertainties in the input data the results should still be interpreted with caution.” (Line 564-573)

Thank you very much again for your comments. They helped us to reflect more on the single paragraphs of our paper and improved our article.

Kind regards

Reviewer 2 Report

The following questions or comments need to be clarified or revised before accept.

(1)    In ‘Materials and Methods’, add a technical roadmap to determine the burden of disease and give corresponding calculation equations on DALYs, YLLs, YLDs and PAF to clarify the calculation method.

(2)    What the physical meaning of ‘the counterfactual value’?

(3)    How does the highly sleep (%SD) be defined.

(4)    Line 112-116, the sentences, for ischemic heart disease mortality attributable to railway noise no exposure-response function is provided by Kempen et al. For some estimates, e. g. aircraft noise and ischemic heart disease morbidity the confidence interval includes one, should be rewritten.

(5)    In the caption of Table 8 in line 278, a redundant bracket should be deleted.

(6)    In the table 10, what is S4 and S10?

(7)    In the sentence of line 42, the WHO defined five of the analyzed health outcomes as critical……, check the usage of ‘critical’.

(8)    In the sentence in line 75-76, this percentage is than applied to the people exposed to different noise levels……, check the word ‘than’.

Author Response

Dear Reviewer,

Thank you very much for your valuable comments. We carefully considered them and addressed them point by point:

(1)    In ‘Materials and Methods’, add a technical roadmap to determine the burden of disease and give corresponding calculation equations on DALYs, YLLs, YLDs and PAF to clarify the calculation method.

Answer: Thank you very much for your comment. We also think, that a technical roadmap will help readers to understand the concept and the calculations. We thus, graphically ordered the burden of disease quantification in a three step process including the calculation equations:

Figure 1: Technical roadmap of the burden of disease quantification referring to [20,21]; YLLs = years of life lost due to premature mortality; YLDs = years lived with disability; DALYs = disability-adjusted life years; PAF= population attributable fraction; RR = relative risk, DALYsN = DALYs attributable to noise

(2)    What the physical meaning of ‘the counterfactual value’?

Answer: The counterfactual value is a crucial part of an environmental burden of disease estimation. It is an alternative scenario, which can have different levels, such as no exposure or a limit value. In the article the counterfactual value is described as:

“To quantify the PAF the number of people exposed to noise, an exposure-response function or relative risks and a counterfactual value (alternative scenario, indicating a minimum or no risk exposure level) are needed [18,19]. The relative risks often include a 95 % confidence interval which is used to quantify the confidence interval for the DALYs, YLLs and YLDs. In our case the counterfactual value gives in our case the starting point for the risk increase of the exposure-response function. Below this value no negative health effects are quantified.“ (Line 67-72)

(3)    How does the highly sleep (%SD) be defined.

Answer: Thank you very much for this comment. We have specified the definition of high sleep disturbance in the article:

“Sleep disturbance can be assessed in many different ways. For this assessment we rely on the WHO recommendations and assess the burden measured by self-reported sleep-disturbance, which is defined as remembering events of awakening from sleep and the disturbed process of falling asleep. If individuals define their sleep as “very” or “extremely” disturbed they are classified as highly sleep disturbed. [1].“ (Line 96-101)

(4)    Line 112-116, the sentences, for ischemic heart disease mortality attributable to railway noise no exposure-response function is provided by Kempen et al. For some estimates, e. g. aircraft noise and ischemic heart disease morbidity the confidence interval includes one, should be rewritten.

Answer: We rewrote the two sentence as followed:

“For ischemic heart disease mortality attributable to railway noise no sufficient evidence is available to derive an no exposure-response function is provided by Kempen et al. [7]. All confidence intervals of the exposure-response functions except for ischemic heart disease morbidity and road traffic noise include one. For some estimates, e. g. aircraft noise and ischemic heart disease morbidity the confidence interval includes one.” (Line 121-126)

(5)    In the caption of Table 8 in line 278, a redundant bracket should be deleted.

Answer: Thank you very much. We deleted the bracket.

(6)    In the table 10, what is S4 and S10?

Answer: S stands for Scenario and the number simply indicates the number of each scenario. The scenarios are explained in table 5 of the method section and the paragraph “sensitivity analysis”.

(7)    In the sentence of line 42, the WHO defined five of the analyzed health outcomes as critical……, check the usage of ‘critical’.

Answer: The word critical was used in this context by the WHO (WHO 2018. Environmental Noise Guidelines for the European Region. p. 10 and following):

As a first step, the GDG identified key health outcomes associated with environmental noise. Next, it rated the relevance of these health outcomes according to the following three categories:

critical for assessing environmental noise issues

important, but not critical for assessing environmental noise issues

unimportant.“

Therefore we would like to keep this wording.

(8)    In the sentence in line 75-76, this percentage is than applied to the people exposed to different noise levels……, check the word ‘than’.

Answer: Thank you very much for point out this misspelling. We corrected the wording. 

Thank you very much again for your comments. They helped a lot to improve our article.

Kind regards

Round 2

Reviewer 1 Report

Some points have been clarified

Reviewer 2 Report

I do not have  additional comments.